# A Study on the Impact of Industrial Restructuring on Carbon Dioxide Emissions and Scenario Simulation in the Yellow River Basin

**Jianhua Liu [1,2], Tianle Shi [1] and Liangchao Huang [1,3,\*]**

1   School of Management, Zhengzhou University, Zhengzhou 450001, China
2   Yellow River Institute for Ecological Protection & Regional Coordinated Development, Zhengzhou University, Zhengzhou 450001, China
3   Institute of Subsurface Energy Systems, Clausthal University of Technology, 38678 Clausthal-Zellerfeld, Germany
\*   Correspondence: liangchao.huang@tu-clausthal.de

**Abstract:** Based on a detailed analysis of the impact mechanism of industrial restructuring on carbon dioxide emissions in the Yellow River Basin, this paper first calculated the carbon dioxide emission data of 57 prefecture-level cities in the Yellow River Basin from 2009 to 2019 and constructed indicators from two dimensions: the advancement and the rationalization of the industrial structure. Then, the Stochastic Impacts by Regression on Population, Affluence, and Technology (STIRPAT) model was used to empirically analyze the influencing factors of industrial structure adjustments on carbon dioxide emissions in the Yellow River Basin. Consequently, changing carbon dioxide emission trends in the Yellow River Basin under various scenarios were predicted. The research observed the following: (1) the eastern part of the Shandong Peninsula Urban Agglomeration and the Energy Golden Triangle have higher carbon dioxide emissions; (2) the advancement of industrial structures in the Yellow River Basin has a better emission reduction effect than the rationalization of industrial structures; (3) increased foreign investment will lead to an increase in carbon dioxide emissions in the Yellow River Basin, and a "Pollution Refuge Effect" will emerge; (4) accelerated industrial transformations and upgrades, high-quality economic development, and a moderate population growth rate are consistent with future development trends.

**Keywords:** Yellow River Basin; carbon dioxide emissions; rationalization of industrial structure; advancement of industrial structure; impact mechanism; scenario simulation

## 1. Introduction

The accumulated emission of carbon dioxide exacerbated global warming and frequent extreme weather events over time. As the world shifts its focus to carbon peaking and carbon neutrality, the significance of low-carbon economic development gradually become apparent [1–3]. China must modernize its industrial structure and develop a new generation of energy systems to meet the 30·60 dual-carbon goals [4]. Energy plays a significant role in the industrial structure of the Yellow River Basin, which contains a sizeable proportion of energy and chemical industries. There is an apparent conflict between ecological, environmental protection, and high-quality economic development, although the Yellow River Basin's emerging high-knowledge industries are on a small scale [5]. In conclusion, this region is representative of China's efforts in reducing carbon dioxide emissions. In-depth research on upgrading industrial structures in the Yellow River Basin and the realization of green and low-carbon development is crucial and time-sensitive. Critical prerequisites for ecological protection and high-quality development in the Yellow River Basin include the scientific measurements of carbon dioxide emissions, the clarification of the relationship between carbon dioxide emissions and industrial structure, and

the investigation of the impact mechanism of industrial structure adjustment on carbon dioxide emissions.

Previous research demonstrated that the primary determinants of regional carbon dioxide emissions are industrial structure, population, economic growth, urbanization level, and energy intensity [6–8]. Relevant research on the effect of industrial reorganization on carbon dioxide emissions can be divided into three categories. First, one of the standard research designs is to use industrial restructuring as one of the control variables for carbon dioxide emissions and to examine the impact of industrial restructuring on carbon dioxide emissions using input–output models, regression analysis, and other techniques. Most studies have shown that secondary industries have the most significant and consistent positive impact on regional carbon dioxide emissions [9,10]. Meanwhile, some have found that the growth of China's tertiary industries will harm regional carbon dioxide emissions [11,12]. Second, it is a typical research design to decompose carbon dioxide emission factors using a carbon dioxide emissions decomposition analysis approach such as Logarithmic Mean Divisia Index (LMDI). Hence, it examines the impact of industrial restructuring on carbon dioxide emissions; analyzes the relationship between manipulation and results using models such as the Stochastic Impacts by Regression on Population, Affluence, and Technology (STIRPAT) model; and predicts its change pattern. Wang et al. [13] and Pan et al. [14] noted that industrial structure and energy structure are the primary factors influencing industrial carbon dioxide emissions and that industrial structure shifted from contributing to carbon dioxide emission reduction efforts to inhibiting such efforts. Using LMDI model analyses, Fan et al. [15] discovered that industrial restructuring could help reduce carbon dioxide emissions. In addition, the coupling and correlation between carbon dioxide emission and industrial structure in different regions are studied by a grey correlation degree model and coupling coordination degree model, and the results show that the correlation between the secondary industry and carbon emission is the largest [16,17].

In terms of predicting carbon dioxide emissions, the Low Emissions Analysis Platform (LEAP) model and the STIRPAT model are the most prevalently employed techniques in current research studies. In addition, carbon dioxide emission forecasting is supplemented by scenario analysis, which involves designing various scenarios, simulating the changing trends in influencing factors to estimate the carbon peak, and determining the optimal time to reduce carbon dioxide emissions. Emodi et al. [18] predicted the energy demand and carbon dioxide emissions in Nigeria under four different energy policy scenarios by building a LEAP model. Yang et al. [19] analyzed the energy consumption and carbon dioxide emission trends of various industries in China from 2016 to 2050 based on the LEAP model. However, the lack of official data support for industry-related parameters can lead to significant discrepancies between research results and real-world situations when using the LEAP model. In contrast, the STIRPAT model has much room for expansion and improvement and has become increasingly popular in the field of carbon dioxide emission forecasting in recent years. Thio E et al. [20] employed an extended STIRPAT model in combination with panel quantile regression to analyze the driving factors of carbon dioxide emissions across the top 10 countries in the world. Based on the STIRPAT model, Muhammad Khalid Ansern et al. [21] found that fossil fuel consumption, population growth, rising affluence, and urbanization were factors contributing to high carbon dioxide emissions in Pakistan.

Domestic and international scholars established certain research foundations on carbon dioxide emissions and industrial structure. However, the following issues remain. (1) In terms of mechanism analyses, few studies on analyzing the mechanism of industrial restructuring on carbon dioxide emissions have been conducted, and few studies have analyzed industrial structures from the perspectives of rationalization and advancement. (2) In terms of scenario forecast, the impact of industrial restructuring on carbon dioxide emissions has not been explored adequately. (3) Regarding the research scale, relevant studies are more situated in the provincial level and mostly provide a strong macro-level perspective, failing to render effective references for formulating carbon dioxide

emission reduction policies. Consequently, this paper analyzes the mechanism of industrial reorganization in the Yellow River Basin and its effect on carbon dioxide emissions. From 2009 to 2019, the carbon dioxide emissions data of 57 prefecture-level cities in the Yellow River Basin were measured. First, the Kriging interpolation technique carried out a visual analysis of carbon dioxide emissions. Then, using the STIRPAT model, an empirical analysis of influencing factors such as rationalization and advancement of the industrial structure was conducted. Finally, simulations and projections of carbon dioxide emission trends in the Yellow River Basin were conducted under various scenarios to provide scientific decision-making support for industrial restructuring and carbon dioxide emission reduction efforts with respect to the Yellow River Basin.

The innovations of this study are as follows: (1) the emission reduction effect of industrial structure adjustments is discussed from both theoretical and empirical perspectives, which enriches the research on the influencing factors of carbon dioxide emissions and is also a useful supplement relative to the research on industrial structure adjustments; (2) the study was conducted from the perspective of upgrading operations and rationalization to measure the adjustment of industrial structures; (3) combined with the actual data of the Yellow River Basin and the results of regression analysis, eight different scenarios were set to predict carbon dioxide emissions; (4) fifty-seven prefecture-level cities in the Yellow River Basin were selected as the research area. The research scale is more micro-scaled, and the research conclusions are more realistic.

## 2. Theoretical Analysis

Industrial structure refers to the interrelationships between industries and elements. Industrial structure is divided into two dimensions: industrial structure rationalization and industrial structure advancement. Numerous researchers have empirically investigated these two dimensions [22,23]. The rationalization of the industrial structure focuses on the degree of the reasonable resource allocation and their coupling coordination, whereas the advancement of industrial structure seeks to investigate the path of the industrial structure's development from low- to high-level developments, which also represents the direction of the economic development of the service industry and the inclusion of the development of producer services. This paper analyzes how the industrial structure of the Yellow River Basin affects carbon dioxide emissions using this theoretical framework. The mechanism of industrial structure adjustment on carbon dioxide emissions is shown in Figure 1.

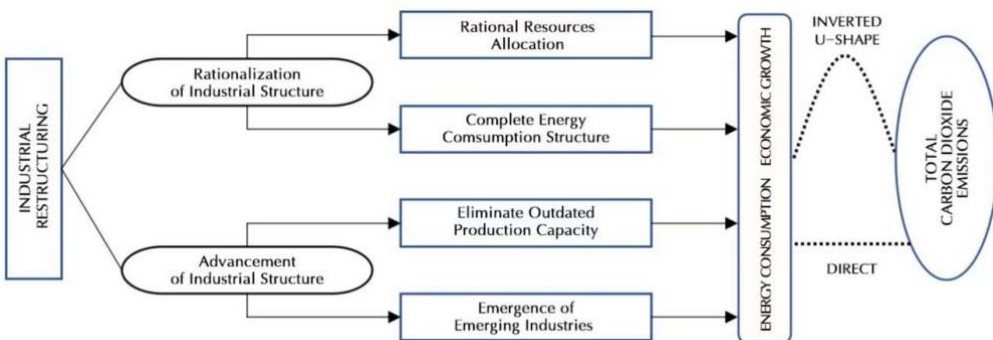

**Figure 1.** Analysis framework for the impact mechanism of industrial structure adjustment on carbon dioxide emissions.

### 2.1. The Impact of Economic Development on Carbon Dioxide Emissions

The adjustment of industrial structure is a significant factor influencing economic development. The transfer of production factors facilitates the modernization and optimization of industrial structures, thereby enhancing and promoting production efficiency and economic development. Therefore, economic development as a medium relates to

the industrial structure of the Yellow River Basin and carbon dioxide emissions from a macro-perspective.

The rationalization of industrial structures is typically accomplished using the allocation of resources rationally, whereas changes in the leading industrial sectors will impact regional economic development. In the early stages of industrialization in the Yellow River Basin, a primary industry dominated the market, and economic development was relatively inefficient. As a result of the expansion of the secondary industry, the industrial sector has gradually taken the lead. The use of advanced production technology and the scale effect of industrial production increased economic benefits. In the meantime, the growth of the energy and chemical industries also contributed to accelerated economic developments. During this period, carbon dioxide emissions also increased significantly. Moreover, when the service sector is dominant, developing industries with the technology, human resources, and knowledge-intensive characteristics will make the allocation of resources more reasonable and greatly improve production efficiencies, which will reduce carbon dioxide emissions.

Accelerating the advancement process of industrial structures will promote the improvement of society's production efficiency and the refinement of the social labor division. It will result in an economic structure that is more service-oriented and shifts it from being a low to a high value-added structure. The combination of industrial structure development and technological advancement resulted in the constant elimination of obsolete production capacities. On the one hand, applying advanced new technology improved the technology intensity of businesses and the residual value of products, leading to a high degree of industrial intensification. In addition to promoting economic growth, this reduces carbon dioxide emissions. On the other hand, developing new industrial sectors drives market growth and increases a society's consumption potential, resulting in larger-scale economic benefits.

Combining the Environmental Kuznets Curve (EKC) theory [24] with the current situation of middle-late stage industrialization in the Yellow River Basin, it is proposed that economic development is becoming less dependent on energy-intensive industries and shifting gradually to greener emerging industries and that economic growth is currently conducive to reducing carbon dioxide emissions.

### 2.2. The Impact of Energy Consumption on Carbon Dioxide Emissions

The rationalization of the industrial structure is reflected in the coordinated development of multiple industries and the efficient use of resources. On the one hand, rationalization increases the energy utilization efficiency of society, eliminates outdated production capacity, and reduces energy waste, reducing energy consumption in product production and carbon dioxide emissions. On the other hand, optimizing the structure of energy consumption would promote the use of clean energy and the development of new energy enterprises. In addition, the rationalization of industrial structure is accompanied by advances in science and technology and equipment replacement. Consequently, energy utilization improved due to minimized energy loss during excavation, processing, storage, and utilization processes, resulting in decreased carbon dioxide unit emissions.

The advancement of industrial structure is accompanied by an increase in the output value of the service sector and a decline in the industry's output value. The workforce from the secondary industry is transferred to the tertiary industry; obsolete production capacity is eliminated; enterprises with high expertise, advanced technical content, and high added value are retained; energy utilization efficiency is enhanced [25]. As a result, economic development will become less reliant on energy. Moreover, sophistication encouraged the development of emerging industrial sectors with high added value and lower energy consumption. At the same time, the emergence of renewable energy and environmentally friendly materials reduced fossil energy use and carbon dioxide emissions due to their "clean characteristics".

## 3. Methodology

The general steps of this research study are as follows: (1) Firstly, 57 prefecture-level cities were selected as research regions and data sources were explained according to the completeness of government planning, administrative divisions, and the correlation between economic and social development and the Yellow River Basin. (2) Secondly, based on the theory of the effect of industrial restructuring on carbon dioxide emissions, eight major energy sources are selected to measure carbon dioxide emissions by referring to the Intergovernmental Panel on Climate Change (IPCC) method. (3) The Kriging model is selected for visually analyzing carbon dioxide emissions, and then the index of rationalization and upgrading of industrial structure is constructed. (4) The STIRPAT model is further expanded, and the influencing factors such as rationalization, upgrading, and urbanization are added.

### 3.1. Research Area and Data Sources

The Yellow River Basin flows through nine provinces, including Qinghai, Sichuan, Gansu, Ningxia, Shaanxi, Henan, Shanxi, Inner Mongolia, and Shandong [26]. Taking government planning, the completeness of the administrative divisions of cities along the Yellow River Basin, and the relationship between economic and social development and the Yellow River Basin as the primary considerations, this paper selected 57 prefecture-level cities where the core areas of the mainstream and major tributaries of the Yellow River Basin are located as the primary research targets. There are 17 prefecture-level cities from the upper reaches in Qinghai, Ningxia, Gansu, and Inner Mongolia [27], 25 from the middle reaches in Shaanxi, Shanxi, and Henan and Gansu provinces; and 15 from the lower reaches in Henan and Shandong provinces. The specific study area is shown in Figure 2. Most data for each indicator come from the *China City Statistical Yearbook* of each city (state), the National Economic and Social Development Statistical Bulletin of each city (state), the bulletin of the relevant department of the environmental protection bureau, the *China Energy Statistical Yearbook*. A portion of the missing data was filled in by averaging or interpolating data from neighboring years.

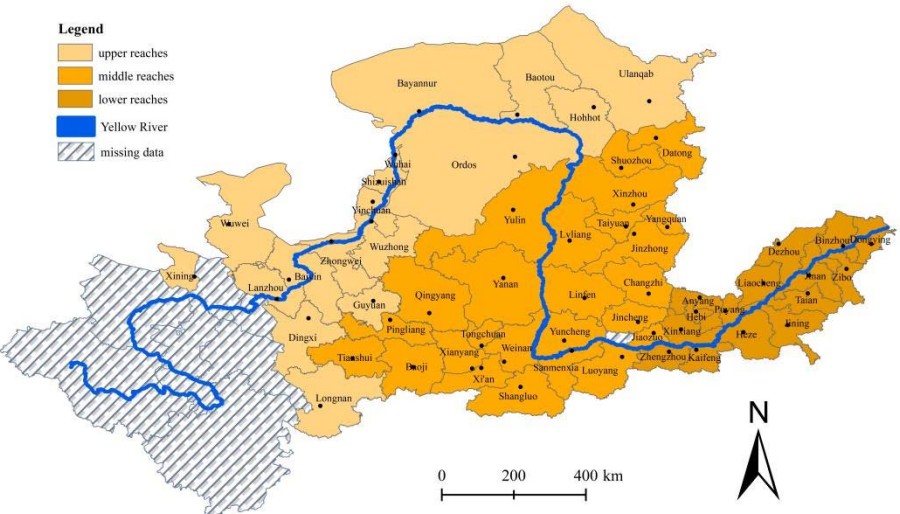

**Figure 2.** The research areas.

### 3.2. Calculation of Carbon Dioxide Emissions

Before investigating the impact mechanism of industrial restructuring on carbon dioxide emissions in the Yellow River Basin, carbon dioxide emissions in this region must be measured. Considering the types and distribution of energy in the Yellow River Basin, this paper refers to the energy consumption data from carbon dioxide emission accounts and datasets (CEADs) and the *China Energy Statistical Yearbook*. This study selects eight

energy sources, such as coal, coke, and crude oil, as the principal energy consumption representatives. The IPCC adopted the following formula to estimate carbon dioxide emissions [28]. The precise method of calculation is as follows:

$$CO_2 = \sum_{i=1}^{8} E_i \times NCV_i \times CC_i \times COF_i \times \frac{44}{12} \tag{1}$$

where $CO_2$ represents the carbon dioxide emissions, $E_i$ represents the consumption of various energy types, $NCV_i$ represents the average low calorific value of the $i$ energy, $CC_i$ represents the carbon content per unit of heat, $COF_i$ represents the oxidation factor of the $i$ energy, and 12 and 44 are the molecular weights of carbon and carbon dioxide, respectively. Table 1 provides energy consumption coefficients.

**Table 1.** Standard Coal Conversion Coefficient and Carbon Emission Coefficient of Different Energy Types.

| Type of Energy | Standard Coal Conversion Coefficient (kgce/kg) | Carbon Emission Coefficient (kgce/kg) |
|---|---|---|
| Coal | 0.714 | 0.756 |
| Coke | 0.971 | 0.855 |
| Crude Oil | 1.483 | 0.590 |
| Gasoline | 1.471 | 0.590 |
| Kerosene | 1.471 | 0.570 |
| Diesel | 1.457 | 0.590 |
| Fuel Oil | 1.429 | 0.620 |
| Natural Gas | 12.143 | 0.448 |
| Electricity | 0.123 | 0.213 |

*3.3. Visualization Model of Carbon Dioxide Emissions*

Kriging interpolation, also known as the spatial interpolation method, is a method that examines regional variables within a limited area based on the variogram theory and structural analyses; it is unbiased and optimal [29]. The Kriging interpolation analyses on the carbon dioxide emissions of prefecture-level cities in the Yellow River Basin can more accurately reflect the distribution of carbon dioxide emissions in the upper, middle, and lower reaches, as well as the environmental impact of each city's carbon dioxide emissions on the surrounding area. It serves as a guide for formulating emission reduction policies. The calculation formula is described as follows:

$$\hat{M}_0 = \sum_{i=1}^{n} \partial_i M_i \tag{2}$$

where $\hat{M}_0$ is the estimated value at point $(x_0, y_0)$, $\partial_i$ is the spatial weight coefficient, $M_i$ is the value of the spatial element, $n$ is the number of cities in the Yellow River Basin, and it represents carbon dioxide emissions at point $i$.

*3.4. Index Construction for Rationalization of Industrial Structure*

Currently, the rationalization level of industrial structure is measured primarily by the structural deviation index; however, this evaluation index has several drawbacks. The Theil index [30] is a modification of this index. However, the logarithmic function magnifies factor consumption, reduces factor scarcity, and disregards absolute values, leading to the "distortion" and "false rationalization" of the industrial structure. All indicators stated above evaluate the rationalization of industrial structures based on a single factor. In contrast, actual economic activities should also consider other factors, such as labor and capital. The distribution of these factors will also have a substantial effect on the industrial structure. In this paper, a weighted structural deviation is used to represent the rationalization of the industrial structure, whereas the employment structure represents

the factor input's structure. In addition, given the normalcy of economic disequilibrium, the proportion of output value is used to determine the relative size of the three largest industries in the economic field. The method of calculation is as follows:

$$RIS = \sum_{w=1}^{n} \left( \frac{Y_{w,t}}{Y_t} \right) \left| \frac{Y_{w,t}/L_{w,t}}{Y_t/L_t} - 1 \right| \tag{3}$$

where *RIS* (rationalization of industrial structure) represents the rationalization level of the industrial structure, $Y_t$ represents the total output (CNY: yuan) at time $t$, $L_t$ represents the employed population (person) at time $t$, $Y_{w,t}$ represents the total output (yuan) of industry $w$ at time $t$, and $L_{w,t}$ represents the employed population (person) in industry $w$ at time $t$. The lower the *RIS* value, the more irrational the industrial structure; the higher the *RIS* value, the more rational the industrial structure.

### 3.5. Index Construction for Advancement of Industrial Structure

Utilizing a ratio of output value, such as the ratio of the output value of the tertiary industry to that of the secondary industry, academia typically evaluates the advancement of an industrial structure. However, this quantitative index does not reflect the progression of the industrial structure from the secondary to the tertiary sector [31]. In addition, the Yellow River Basin is a significant national grain production center, so the primary sector should not be overlooked. In actuality, the advancement of the industrial structure is primarily reflected in two connotations: (1) the evolution trend of the industrial structure from the primary industry to the secondary industry and then to the tertiary industry and (2) the transition of industrial sectors from low value-added products to high value-added products, the movement from the bottom to the top of the value chain. Based on the availability of data, this paper selects an industrial-structure-level index to measure the development of industrial structures. The formula for its calculation is as follows:

$$HIS = \sum_{m=1}^{3} Y_m \times m \tag{4}$$

where *HIS* (high-ranking industrial structure) represents the advancement level of industrial structure, and $Y_m$ represents the proportion of the output value of the $m$ industry in GDP. This index denotes the transition of the dominant position from a primary industry to the secondary and tertiary industries, as well as the goal and direction the industrial structure's upgrades. The larger the *HIS*, the more advanced the industrial structure.

### 3.6. Econometric Model Construction

The IPAT (I = human impact, P = population, A = affluence, and T = technology) identity has the problem of changing at the same rate, so the STIRPAT model must mitigate this problem by predicting the environmental impact when demographic, technological, and economic changes are considered in their entirety [32]. This model is currently the standard method for simulating and predicting carbon dioxide emissions and is highly extensible. This paper adopts the STIRPAT model for empirical research as a result. In conjunction with the factors above, industrial restructuring primarily influences carbon dioxide emissions via the rationalization and advancement of industrial structures. This paper continues the final model using the total amount of carbon dioxide (C) as the dependent variable. The model is constructed as follows:

$$\ln C_{it} = \alpha_0 + \beta_1 \ln RIS_{it} + \beta_2 \ln HIS_{it} + \beta_3 \ln GDP +$$
$$\beta_4 \ln POP + \beta_5 \ln Trade + \beta_6 \ln Urb + \mu_{it} \tag{5}$$

where $i$ represents the city, $t$ represents the year, $C$ represents the carbon dioxide emission, *RIS* represents the rationalization level of the industrial structure, *HIS* represents the advancement level of the industrial structure, *GDP* represents the regional per capita *GDP*,

*POP* represents the regional population, Trade represents the degree of foreign investment as expressed by the degree of foreign investment, *Urb* represents the urbanization rate, $\alpha_0$ represents the constant term, and $\mu_{it}$ represents the random disturbance term.

## 4. Empirical Analysis and Model Testing

### 4.1. Spatial Distribution of Carbon Dioxide Emissions

Using ArcGIS software, this paper employs the Kriging spatial interpolation method to visualize the Yellow River Basin's carbon dioxide emissions in 2009 and 2019. This makes it possible to circumvent the effects of administrative boundaries and more accurately reflect the Yellow River Basin's carbon dioxide emissions. Carbon dioxide emissions in the Yellow River Basin's upper, middle, and lower reaches in 2009 exhibited a "medium-low-high" distribution pattern, as depicted in Figure 3. The lower and upper reaches of the Yellow River Basin, also known as the Energy Golden Triangle and centered on Yinchuan, Ordos, and Yulin, are cities with high carbon dioxide emissions. These densely populated cities are dominated by the energy and chemical industries and are experiencing rapid economic development. On the other hand, most cities with low carbon dioxide emissions are in the middle and upper reaches of the southern Yellow River Basin, where their economic development is relatively underdeveloped. Comparing the Kriging spatial interpolation maps of 2009 and 2019, as depicted in Figure 4, because the Energy Golden Triangle region relies on developing energy-intensive and high-energy-consumption industries, carbon dioxide emissions in the eastern portion of the Shandong Peninsula Urban Agglomeration are relatively high. As a result, the rate of economic expansion is relatively rapid, but it also increases carbon dioxide emissions. Carbon dioxide emission levels in the Lanxi Urban Agglomeration from the upper reaches and the overall cities from the middle reaches continued to decrease, while carbon dioxide emission levels in the Zhengzhou to Jinan section from the lower reaches have also decreased, indicating an improvement in the situation regarding emission reductions.

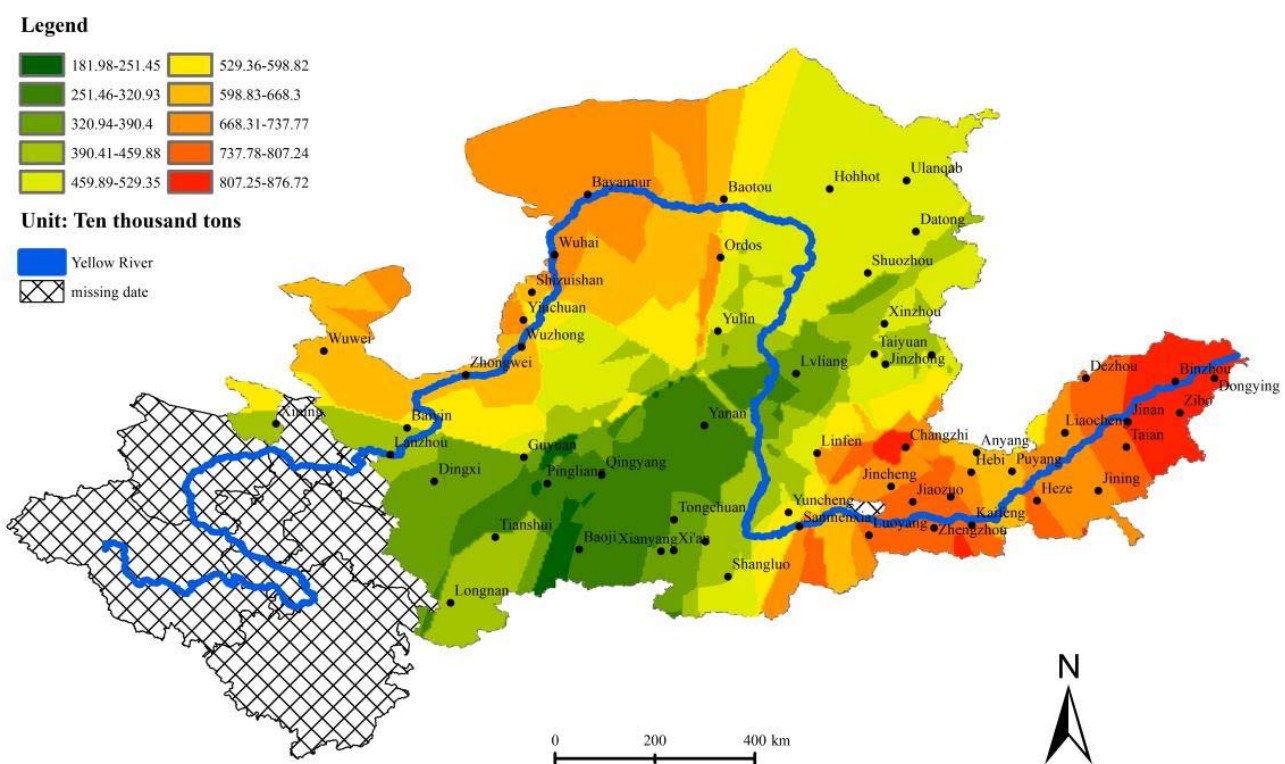

**Figure 3.** Kriging-based spatial interpolation map of carbon dioxide emissions in the Yellow River Basin in 2009.

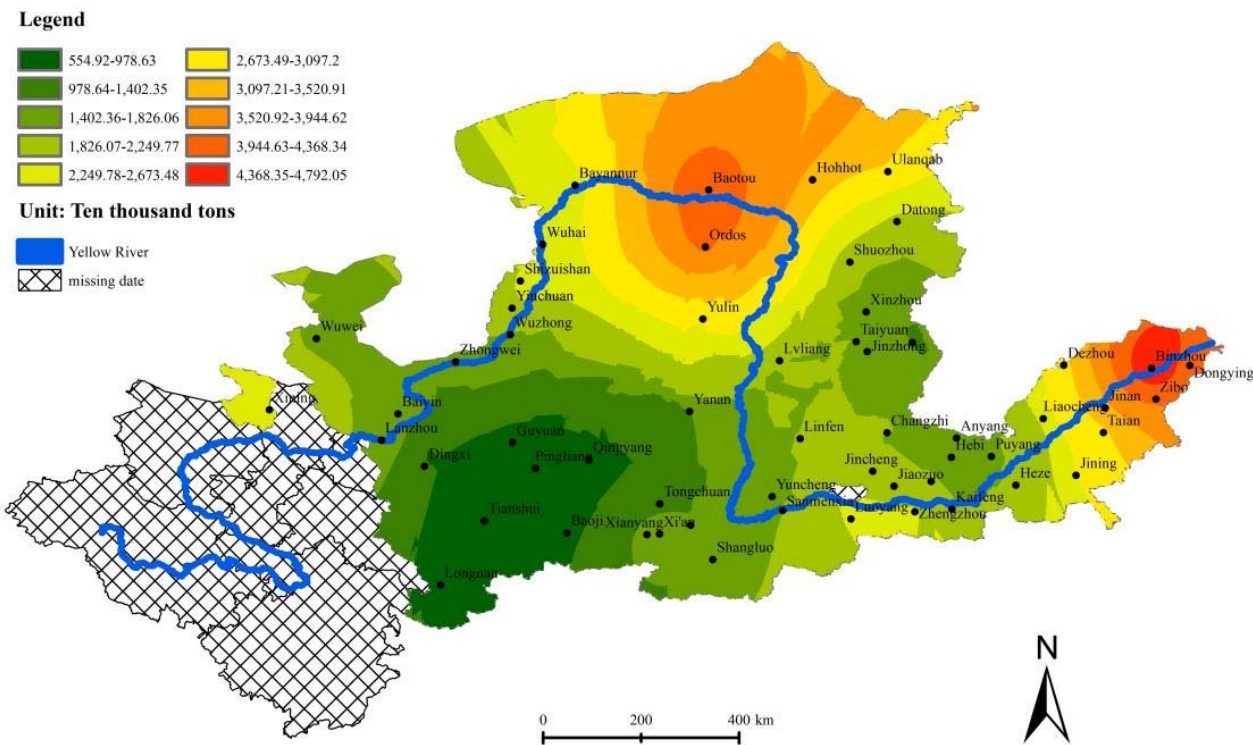

**Figure 4.** Kriging-based spatial interpolation map of carbon dioxide emissions in the Yellow River Basin in 2019.

### 4.2. Panel Model Selection

This paper analyzes the relationship between industrial restructuring and carbon dioxide emissions using Stata 16.0 software. This is performed by first determining whether an individual fixed effects model or an individual random effects model will be constructed. This paper employs the Hausman test and proposes a null hypothesis of constructing a model with individual fixed effects. The alternative hypothesis is to construct a model with individual random effects. The test results indicate that the *p*-values for all models were greater than 0.1, and there was insufficient evidence to reject the null hypothesis; thus, an individual fixed effects model was adopted. Moreover, this study's sample consists of the carbon dioxide emissions and other variables of the 57 prefecture-level cities in the Yellow River Basin, which is not a random sample. Consequently, it is more prudent to select an individual fixed-effect model. Descriptive statistics of variables are shown in Table 2.

**Table 2.** Descriptive statistics of variables.

| Variables | Average | Standard Deviation | Min. | Max. |
|---|---|---|---|---|
| Carbon dioxide emissions (ten thousand tons) | 3279.634 | 1845.193 | 464.487 | 10,847.950 |
| Rationalization of Industrial Structure | 4.377 | 11.166 | 0.067 | 113.410 |
| Advancement of Industrial Structure | 2.292 | 0.133 | 2.039 | 2.650 |
| GDP per capita (ten thousand CNY) | 4.821 | 3.472 | 0.449 | 25.688 |
| Population (ten thousand people) | 357.039 | 218.220 | 18.100 | 1072.500 |
| Urbanization Rate (%) | 42.233 | 26.960 | 0.0440 | 95.000 |
| Degree of Foreign Investment | 1.155 | 1.403 | 0.0174 | 7.005 |

### 4.3. Analysis of Regression Results

In this paper, regression on the panel data was performed using the Stata 16.0 software, and the results are presented in Table 3. Column (2) adds variable controls based on column (1). From column (2), the weighted structural deviation of the index that measures

the level of rationalization in the industrial structure passed the 5% significance test and is positively correlated with carbon dioxide emissions. Due to the negative index of the weighted structural deviation, the higher the value, the more irrational the industrial structure, indicating that a rational industrial structure can effectively reduce carbon dioxide emissions. When the weighted structural deviation increases by 1%, the irrational level of the industrial structure increases by 1%, and carbon dioxide emissions increase by 0.0286%. Therefore, the weighted structural deviation reduced; the rationalization of the industrial structure improved; the utilization of resources is more rational and complete; the level of coupling between the industry and employment structure increased; energy usage efficiency is enhanced, thereby reducing carbon dioxide emissions.

**Table 3.** Results of benchmark regression analysis.

|  | **(1)** | **(2)** |
|---|---|---|
| Model | Fe | Fe |
| Variable | lnC | lnC |
| lnRIS | 0.0249 ** | 0.0286 ** |
|  | (0.00965) | (0.0135) |
| lnHIS | −1.277 *** | −0.742 *** |
|  | (0.205) | (0.262) |
| lnGDP | 0.315 *** | 0.210 *** |
|  | (0.0144) | (0.0212) |
| lnPOP | 0.328 *** | 0.203 * |
|  | (0.0964) | (0.106) |
| lnUrb |  | −0.0558 |
|  |  | (0.0454) |
| lnTrade |  | 0.0595 *** |
|  |  | (0.0129) |
| Intercept | 10.07 *** | 9.416 *** |
|  | (0.211) | (0.278) |
| Observed Value | 627 | 627 |
| $R^2$ | 0.67 | 0.501 |

*** $p < 0.01$, ** $p < 0.05$, * $p < 0.1$.

From column (2), the industrial structure level index, which represents the level of the development of an industrial structure, passed the 1% significance test and was negatively correlated with carbon dioxide emissions, indicating that industrial structure developments had a significant inhibitory effect on carbon dioxide emissions. When the industrial structure level index rises by 1%, carbon dioxide emissions fall by 0.742%. Improving the industrial structure level index will simultaneously eliminate obsolete production capacity and encourage the transformation and modernization of businesses. In addition, the proportion of carbon dioxide emissions in the service sector is considerably lower than in the industrial sector, effectively reducing overall carbon dioxide emissions.

Since the coefficient of industrial structure rationalization is less than that of industrial structure advancements, the effect of industrial structure advancements on emission reduction is greater than that of rationalization. The population coefficient is positive and statistically significant, indicating a positive influence on carbon dioxide emissions. The regression coefficient for the level of foreign investment is significant at 1%, indicating that the level of foreign investment has a more significant impact on carbon dioxide emissions in the Yellow River Basin. This is due to the Yellow River Basin's abundance of natural resources and its lower trade openness compared to coastal cities. Therefore, foreign investors in this region are more likely to invest in resource-intensive industries. At the same time, coastal cities that migrated to the Yellow River Basin exclude foreign-funded enterprises with high energy consumption and high pollution levels. In addition, the coefficients of GDP per capita in columns (1) and (2) are both positive and statistically significant at 1%, indicating that economic growth during the period from 2009 to 2019 is

a significant development with respect to the increase in carbon dioxide emissions in the Yellow River Basin.

*4.4. Robustness and Accuracy Test*

This paper examines the following three aspects to test the robustness of the research findings on the impact of industrial restructuring on carbon dioxide emissions and the accuracy of the prediction models.

(1) Sample manipulation: In this study, the winsorization method removes outliers, and new samples are utilized to estimate Equation (3). In column (1) of Table 4, the regression results indicate that the coefficients of rationalization and advancement of the industrial structure are significant at the 5% and 1% levels, respectively, and that the significance and estimated coefficients have not changed significantly from column (2) of Table 3.

**Table 4.** Estimated results of robustness tests.

|  | **(1)** | **(2)** | **(3)** |
|---|---|---|---|
| Model | Fe | Fe | Fe |
| Variables | lnC | lnC | lnC |
| lnRIS | 0.0289 ** | 0.0259 * |  |
|  | (0.0135) | (0.0137) |  |
| lnHIS | −0.752 *** |  | −0.707 *** |
|  | (0.266) |  | (0.267) |
| lnG | 0.210 *** | 0.200 *** | 0.216 *** |
|  | (0.0211) | (0.0210) | (0.0211) |
| lnP | 0.201 * | 0.268 ** | 0.244 ** |
|  | (0.107) | (0.105) | (0.105) |
| lnUrb | −0.0582 | −0.126 *** | −0.0436 |
|  | (0.0457) | (0.0393) | (0.0455) |
| lnTrade | 0.0596 *** | 0.0610 *** | 0.0598 *** |
|  | (0.0129) | (0.0131) | (0.0130) |
| Intercept | 9.420 *** | 8.741 *** | 9.481 *** |
|  | (0.280) | (0.146) | (0.280) |
| Observed Value | 627 | 627 | 627 |
| $R^2$ | 31 | 0.487 | 31 |

*** $p < 0.01$, ** $p < 0.05$, * $p < 0.1$.

(2) Controlling core variables: The industrial structure's rationalization and advancement indexes were successively added to the model to test the model's robustness. The regression results are shown in columns (2) and (3) of Table 4, where the rationalization and advancement coefficients generated after controlling the core variables are comparable and statistically significant relative to columns (1) and (2) of Table 3, indicating that the benchmark regression results are reliable.

(3) Accuracy test of the prediction model: This paper simulates the carbon dioxide emission values from 2009 to 2019 based on the results of the panel data regression and compares the simulated values with historical data, thereby validating the accuracy of the prediction model. As depicted in Figure 5, the deviations between 2009 and 2019 are minimal, with maximum deviations not exceeding 6%. Overall, the trends in the predicted values are consistent with the trends in the actual values, indicating that the model is accurate and that the framework for carbon dioxide emission prediction constructed in this paper is rational.

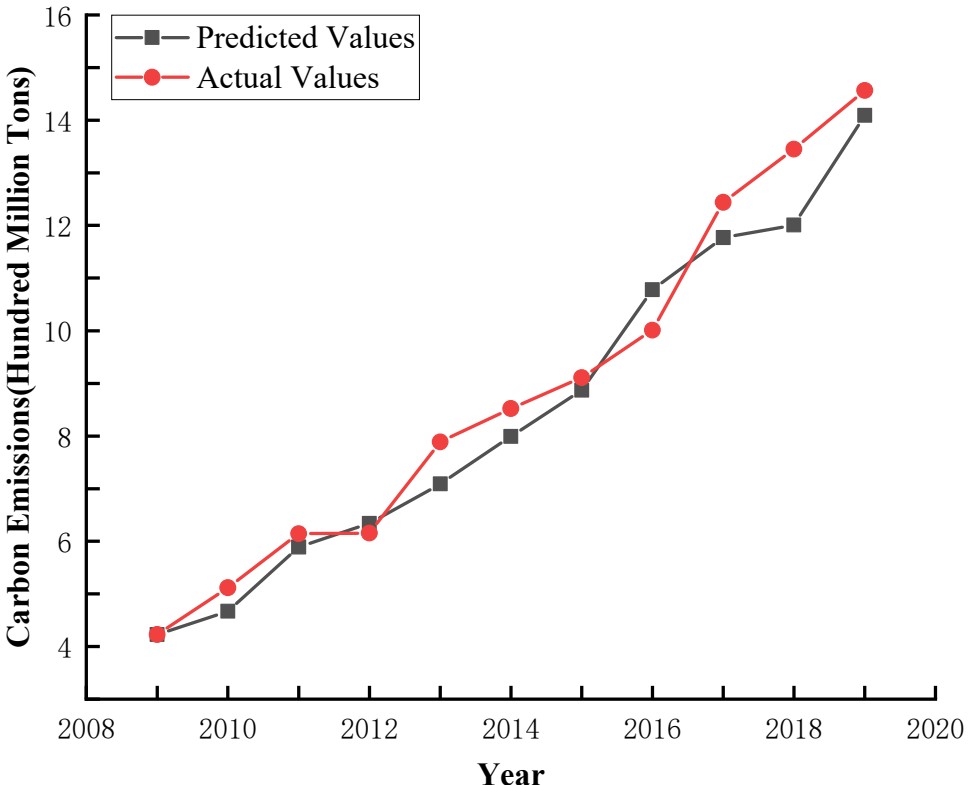

**Figure 5.** Comparison between the actual and predicted carbon dioxide emissions in the Yellow River Basin.

## 5. Scenario Simulation Based on the STIRPAT Model

### 5.1. Scenario Setting

Based on the historical data of carbon dioxide emissions in the Yellow River Basin from 2009 to 2019, the actual development situation, reasonable parameters, and scenarios have been determined. Therefore, this paper assumes that each variable in the STIRPAT model has two possible values: high and low. In the meantime, industrial restructuring comprises the rationalization and advancement of industrial structures; population size comprises population and urbanization rate; the economic level comprises per capita GDP and foreign investment. Based on this, eight scene modes are set according to Table 5.

**Table 5.** Scenario settings.

| Scenarios | Industrial Restructuring | Population Size | Economic Level |
| --- | --- | --- | --- |
| Scenario 1 | Low | Low | Low |
| Scenario 2 | High | Low | Low |
| Scenario 3 | High | High | Low |
| Scenario 4 | High | High | High |
| Scenario 5 | Low | Low | High |
| Scenario 6 | Low | High | High |
| Scenario 7 | Low | High | Low |
| Scenario 8 | High | Low | High |

As this study is based on Yellow River Basin data from 2009 to 2019, the forecast base year is set to 2019, and the forecast year is set to 2035 based on the various scenarios presented in Table 5. The particular parameters are detailed in Table 6.

**Table 6.** Parameter settings under different scenarios.

| Variables | Scenario Parameter Settings | 2020–2025 | 2026–2030 | 2030–2035 |
|---|---|---|---|---|
| Rationalization of Industrial Structure | Low | −0.3 | −0.2 | −0.1 |
| | High | −0.5 | −0.35 | −0.15 |
| Advancement of Industrial Structure | Low | 0.4 | 0.25 | 0.1 |
| | High | 0.6 | 0.45 | 0.2 |
| Population | Low | 0.4 | 0.2 | 0.1 |
| | High | 0.5 | 0.25 | 0.1 |
| Urbanization Rate | Low | 1.6 | 1.2 | 0.8 |
| | High | 1.8 | 1.5 | 1.2 |
| GDP per capita | Low | 5.1 | 3.9 | 2.8 |
| | High | 6.1 | 5.1 | 3.5 |
| Foreign Investment | Low | 1.2 | 0.8 | 0.4 |
| | High | 1.8 | 1.3 | 0.7 |

*5.2. Analysis of Simulation Prediction Results*

As depicted in Figure 6, there are significant differences in the trends and rates of change for carbon dioxide emissions in the Yellow River Basin among the eight scenarios. After 2027, carbon dioxide emissions for Scenario 1, Scenario 2, Scenario 5, and Scenario 8 will decrease steadily, while emissions for all other scenarios will continue to rise. The primary cause of this phenomenon is the rate of population growth. Scenario 7 has the highest rate of increased carbon dioxide emissions because population growth increases energy demands. There has been less industrial restructuring, thereby slowing the optimization of the energy structure and limiting the improvement in overall energy efficiencies. Meanwhile, Scenario 8 has a significantly lower rate of carbon dioxide emission growth than the other scenarios. Under this scenario, against the backdrop of high-quality economic development and stable population growth, the most optimal future development path for the Yellow River Basin is the transformation and modernization of the industrial structure by optimizing the energy structure and innovating energy-saving technologies. However, the Yellow River Basin must implement the most recent national population policy, so industrial restructuring should focus efforts on reducing carbon dioxide emissions while also considering economic development, energy conservation, and emission reduction.

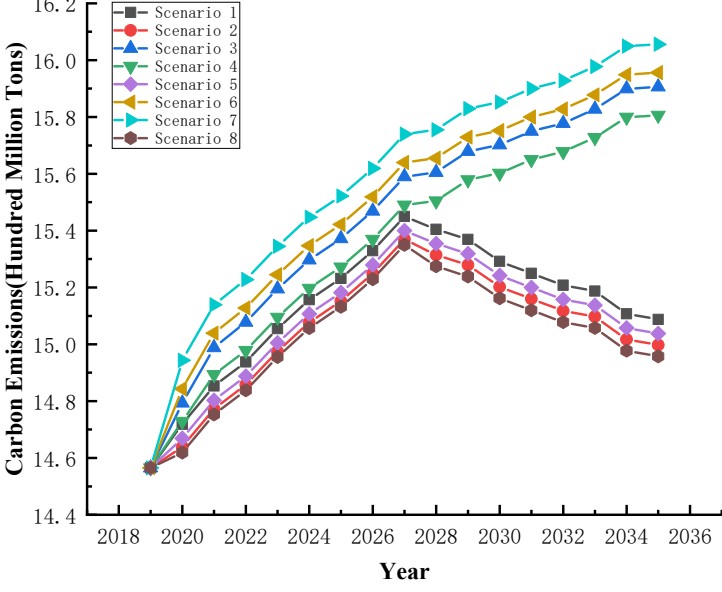

**Figure 6.** Carbon dioxide emission trends in the Yellow River Basin under different scenarios.

## 6. Conclusions and Suggestions

This paper examines how industrial restructuring affects carbon dioxide emissions in the Yellow River Basin. First, it calculates the carbon dioxide emissions data of 57 prefecture-level cities in the Yellow River Basin from 2009 to 2019, performs a visual analysis, and develops an extended STIRPAT model to analyze the effect of industrial restructuring on carbon dioxide emissions. Based on this information, eight distinct development scenarios are created to simulate and forecast future carbon dioxide emission trends in the Yellow River Basin. The main conclusions are provided below.

First, the carbon dioxide emissions in the eastern portion of the Shandong Peninsula Urban Agglomeration and the Energy Golden Triangle are relatively high; these are the most carbon-intensive regions.

Second, the advancement of the industrial structure in the Yellow River Basin has a more significant impact on reducing carbon dioxide emissions than the rationalization of industrial structure. In addition to the rational allocation of production factors, pursuing low-carbon development should focus on developing strategic emerging and high-quality service industries.

Third, the increase in foreign investments will increase the Yellow River Basin's carbon dioxide emissions. Foreign investors are more likely to invest in energy-intensive industries in the Yellow River Basin, and resource-intensive foreign firms are more likely to migrate to the Yellow River Basin, a phenomenon known as the "pollution refugee."

Fourth, the changes in the Yellow River Basin's carbon dioxide emissions vary depending on the scenario. Future development trends include accelerating industrial transformation and modernization, stabilizing population growth, and achieving high-quality economic development.

This paper makes corresponding recommendations regarding the Yellow River Basin's carbon dioxide emissions under various scenarios.

First, the critical areas of carbon dioxide emissions should be based on regional economic development, and industrial restructuring should be accelerated. In future, the eastern portion of the Shandong Peninsula Urban Agglomeration and the Energy Golden Triangle region should change their traditional development model by creating a low-carbon and intensive industrial cluster, promoting the use of renewable energy and gradually reducing their reliance on energy and chemical industries for economic development.

Second, the market and government must play a more prominent role in promoting the rational development of the industrial structure. They need to maintain a favorable market environment, optimize industrial structures, and encourage rational resource allocation, fostering economic development and reducing carbon dioxide emissions. Specifically, the government should actively regulate, guide, and promote the optimization of industrial layouts and high-quality developments in the Yellow River Basin by applying policies such as subsidies, taxation, procurement, and other measures based on the role of market regulation.

Third, the level of science and technology must be increased, while the advancement of industrial structures should be encouraged. Enhancing scientific and technological levels can increase production efficiency and advance high-tech industries. Therefore, it is necessary to promote the development of secondary industries towards low energy consumption and high efficiency and to redirect surplus resources to the tertiary industry [33]. Other suggested methods include improving the level of economic service sectors in the city and leading the industrial structure of the city in achieving advanced developments, eventually reducing the city's carbon dioxide emissions.

Fourth, service industries must be actively relocated to coastal regions, while international investment in industries with low energy consumption and high output performance should be encouraged. Compared to coastal regions, the economic foundation of the Yellow River Basin is relatively weak. Therefore, the government should guide and regulate foreign investment behavior, manage "double high" industries, and encourage the conversion of excess production capacities into the service sectors.

**Author Contributions:** Data curation, methodology, software, visualization, and writing—original draft, J.L. and T.S.; conceptualization, funding acquisition and project administration, and supervision, J.L.; investigation, J.L. and L.H.; resources, T.S. and L.H.; writing—review and editing, J.L. All authors have read and agreed to the published version of the manuscript.

**Funding:** This research was funded by the National Social Science Foundation of China(21FGLB092), the Henan Province Soft Science Major Project (212400410002), and China Scholarship Council (202208080109).

**Institutional Review Board Statement:** Not applicable.

**Informed Consent Statement:** Not applicable.

**Data Availability Statement:** Not applicable.

**Conflicts of Interest:** The authors declare no conflict of interest.

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
