# Peer review of "A Study on the Impact of Industrial Restructuring on Carbon Dioxide Emissions and Scenario Simulation in the Yellow River Basin"

_water, doi:10.3390/w14233833_

Round 1

Reviewer 1 Report

The Yellow River Basin is the second largest river in China. This paper combines the Yellow River Basin with the carbon neutrality strategy to explore the impact of industrial restructuring on carbon emissions in the Yellow River Basin. The topic selection of the article is relatively novel, the conclusion is clear, and it is in line with the research direction of the journal. Some minor questions:

(1)In the introduction part, supplement the relevant literature of environmental economic geography, such as the research content of industrial structure adjustment on ecological environment and carbon emissions.

(2)Enrich the connotation of advanced industrial structure and rationalization of industrial structure, such as the inclusion of the development of producer services into the advanced industrial structure.

 (3) Increase the theoretical analysis of economic growth and energy consumption, such as "reasonable allocation of resources, improvement of energy consumption structure, elimination of backward production capacity, and emergence of emerging industries".

(4)Standardize some text expressions and language structures.

(5)  Some relative references should be added in the section of Introduction and Discussion.

The dominant driving factors of rocky desertification and their variations in typical mountainous karst areas of Southwest China in the context of global change

A novel-optimal monitoring index of rocky desertification based on feature space model and red edge indices that derived from sentinel-2 MSI

The Changes of Spatiotemporal Pattern of Rocky Desertification and Its Dominant Driving Factors in Typical Karst Mountainous Areas under the Background of Global Change

Author Response

Dear reviewer.

I have replied to your valuable comments and revised my paper.Please see the attachment.

Reviewer 2 Report

Review on Paper A Study on the Impact of Industrial Restructuring on Carbon Emissions and Scenario Simulation in the Yellow River Basin

Plagiarism – 29 %

Review:

Introduction part

  • Mention Researchers' contributions through citation instead of vocabulary like" Some scholars" and" Some researchers".
  • Mention the case studies of Industrial restructuring in the issues listed in the paper.

          Research Design, Empirical Analysis & Model Testing

·         Visualization of Maps should follow uniformity like layout, direction placement, outline, and Representation of ROI (Region of Interest).

Calculation of the Carbon emissions,

·         CO2 represent carbon dioxide emissions, not carbon emissions.

Visualisation model

·         Kriging Model formula, “n” parameter is unknown.

Econometric Model

·         Elaborate "IPAT".

Author Response

Dear reviewer

I have replied to your valuable comments and revised my paper.Please see the attachment.

Reviewer 3 Report

Manuscript ID water-2043961 (A Study on the Impact of Industrial Restructuring on Carbon Emissions and Scenario Simulation in the Yellow River Basin) is of great relevance in the context of global climate change due to the approach that emphasizes the importance of population growth and factors industrial development, which imply consumption and, above all, energy consumption, in the provinces along this river of great importance for primary and secondary economic activities in 57 prefecture-level cities in the Yellow River Basin from 2009 to 2019. These factors that were considered for the modeling are, in fact, the ones that most impact GHG emissions, consequently, the increase in temperature and global climate change.

The STIRPAT model was used to empirically analyze the influencing factors of industrial structure adjustment on carbon emissions in the Yellow River Basin. Consequently, the changing carbon emission trends in the Yellow River Basin under various scenarios were predicted. The research found that: (1) the eastern part of the Shandong Peninsula Urban Agglomeration and the Energy Golden Triangle have higher carbon emissions; (2) the advancement of industrial structures in the Yellow River Basin has a better emission reduction effect than the rationalization of industrial structures; (3) increased foreign investment will lead to an increase in carbon emissions in the Yellow River Basin, and a "Pollution Refuge Effect" will emerge; and (4) accelerated industrial trans formation and upgrading, high-quality economic development, and a moderate population growth rate are consistent with future development trends.

The STIRPAT model fitted well with simulation of 8 Scenarios based on the Economic Level of Industrial Restructuring of Population Size for Carbon Emission Trends in the Yellow River Basin.

However, some changes are necessary, as in item 2. Analysis Methodology, which extends extensively in a theoretical framework, which does not fit the methodology, but in a synthetic way in the 1. Introduction.

This item should be focused on the description in Figure 1.

Item 2. Methodology Analysis must include:

3. Research Design 3.1 Research area and data sources; 3.2 Calculation of carbon emissions; 3.3 Visualization model of carbon emissions; 3.4 Index construction for rationalization of industrial structure; 3.5 Index construction for advancement of industrial structure; 3.6 Econometric model construction;

Therefore, it is recommended to publish the article after these minor corrections.

Author Response

Dear reviewer

I have replied to your valuable comments and revised my paper. Please see the attachment.

Round 2

Reviewer 2 Report

The Author had revised based on revisions submitted during the First review.